# Nonlinear Sorption of Organic Contaminant during Two-Dimensional Transport in Saturated Sand

**Sang-Gil Lee [1], Soonjae Lee [1,\*] and Jae-Woo Choi [2,\*]**

1 Department of Earth and Environmental Sciences, Korea University, Anam-ro 145, Sungbuk-gu, Seoul 02841, Korea; sg0403@korea.ac.kr
2 Center for Water Resources Cycle Research, Korea Institute of Science and Technology, Hwarang-ro 14-gil 5, Seongbuk-gu, Seoul 02792, Korea
\* Correspondence: soonjam@korea.ac.kr (S.L.); plead36@kist.re.kr (J.-W.C.); Tel.: +82-2-3290-3177 (S.L.); +82-2-958-5820 (J.-W.C.)

**Abstract:** Multi-dimensional transport studies are necessary in order to better explain the fate of contaminants in groundwater. In this study, a two-dimensional transport experiment with organic contaminants in saturated sand was conducted to investigate the migration of the organic contaminant plume in multi-dimensional flow conditions. The transport test was conducted using toluene as a model organic contaminant in a saturated sand box under steady flow conditions. The initial plume was generated via injection at a point source. After 24 h, the plume distribution was delineated by interpolating toluene concentrations in the porewater samples. The mass centers of the toluene and the conservative tracer were almost coincident, but the size of the toluene plume was significantly reduced in longitudinal as well as transversal directions. The appropriateness of several types of sorption models were compared to describe the toluene sorption in two-dimensional transport system using numerical modeling. Among the sorption models, the Langmuir model was found to be the most appropriate to describe the sorption of toluene during two-dimensional transport. The results showed that two-dimensional experiments are better than one-dimensional column experiments in identifying the adsorption characteristics that occur during transport in saturated aquifers.

**Keywords:** 2D sand box; toluene; transport; sorption; reversibility; local equilibrium assumption; non-linear isotherm

## 1. Introduction

The leakage of petroleum oil from facilities designed for transport and storage is one of the major sources of groundwater contamination. Leaked oil penetrates into the underground environment and forms a non-aqueous liquid (NAPL) pool above or below the aquifer. Various contaminants are continuously supplied into the groundwater through the dissolution of organic compounds from the NAPL pool. The presence of organic contaminants in the groundwater could pose a serious hazard to public health and the environment. For example, one of the most common hydrocarbon contaminants in soil and groundwater is BTEX (benzene, toluene, ethylbenzene, xylene) [1]. BTEX compounds are more toxic than liquid alkanes and are well-known toxicants to a wide range of biota [2]. Benzene is an International Agency for Research on Cancer (IARC)-classified group I carcinogen, ethylbenzene is a suspected IIB carcinogen, and both toluene and xylenes are IARC group III neurotoxins [3]. Groundwater contaminated by BTEX is a very serious problem because many communities in the world depend upon groundwater as a sole or major source of drinking water [4]. USEPA reported that the maximum levels for monoaromatic compounds in potable water are 0.05, 1, 0.7 and 10 mg/L for benzene, toluene, ethylbenzene and isomers of xylenes, respectively [5].

Among these, toluene is used in various fields for many purposes, such as a solvents in paint, lacquers, thinner, glue and nail polish, as well as in leather tanning [6,7]. It is

also used as an octane booster in gasoline fuels in internal combustion engines, and in cosmetic and personal care products [8–10]. Therefore, toluene has been reported as a potential substance which can contaminate groundwater and soil. In particular, toluene is known to be a degradable material in the underground environment, but analysis of the transport characteristics of toluene is very important, since the removal rate can be changed, depending on environmental conditions. It is necessary to confirm whether the natural attenuation process can be degraded based on the prediction of the behavior of toluene when underground environmental pollution occurs due to an intensified toluene spill [11].

The fate of organic contaminants in groundwater is controlled by advective and dispersive mass transfer, as well as attenuation associated with sorption [12]. The sorption processes in the subsurface environment are very complex, often involving non-linear phase relationships and rate-limited conditions [13]. The sorption of an organic compound on the aquifer material affects the distribution of contaminants and is important in understanding the fate of the contaminant in the aquifer. The adsorption of organic compounds onto the aquifer material is highly dependent on the organic matter in the soil ($f_{oc}$), as well as the surface area related to the clay contents [13,14]. Because the critical level of soil organic matter is low (0.1% for the case of benzene), the adsorption of organic contaminant onto the aquifer is mostly dominated by the hydrophobic reaction of soil organic matter. In field conditions, heterogeneous soil properties, such as organic matter, further complicate the fate of organic contaminants [15–17].

Hydrophobic adsorption studies have been conducted on organic pollutants in soil under various conditions [18]. The adsorption reaction between clay minerals and organic substances in the soil causes adsorption, leading to heterogeneous adsorbents [19]. The distribution of pollutants in soil and groundwater can be explained by non-linear forms of the Langmuir and Freundlich isotherm models [20]. The Langmuir sorption model estimates monolayer adsorption for adsorbents, whereas the Freundlich model is commonly applicable to heterogeneous sorption [21]. The characteristics of non-linear sorption do not appear when the concentration of organic pollutants is low; therefore, it can be explained with a linear sorption isotherm [22].

Numerous experimental and modeling studies have been performed at laboratory- and/or field-scales to examine the effect of the sorption of organic contaminants onto aquifer materials and its effects on their transport phenomena [14,23–33]. In particular, column experiments have been frequently used to investigate the fate of organic contaminants [26]. Several laboratory column studies have shown the attenuation of organic contaminants during transport [27–29]. The retardation of organic pollutants has also been reported in one-dimensional column experiments, as well as in the two-dimensional sand box test [30–32]. In several field studies, highly retarded transport trends were reported for hydrophobic compounds [33]. Long-term experiments at the site scale (Borden, Ontario) confirmed the occurrence of retardation, along with attenuation, and this behavior could be successfully explained using the linear equilibrium approach [14].

The equilibrium approach using a partitioning coefficient ($K_d = f_{oc} \times K_{oc}$) is widely used to describe sorption during the transport of organic contaminants in aquifers [14,30–32]. However, the linear sorption property is not applicable in all aspects. Previous studies have reported on the differences in the retardation caused by non-linear and linear sorption using a one-dimensional column [34,35]. When the content of organic matter in the soil is low, the functional groups in soil which can react with pollutants may be saturated within a limited time. In this case, it may appear that irreversible sorption has occurred on the soil surface during the transport of pollutants due to the termination of the sorption reaction because of saturation. To confirm this reaction, it is necessary to conduct a test on the transport of pollutants through a multi-dimensional laborator-scale model experimental apparatus. We thought that the two-dimensional transport test could reflect the transport characteristics at the contaminated site better than the one-dimensional test as a method of the laboratory test.

The objective of this study was to investigate the type of sorption that occurred during the transport of an organic contaminant in two-dimensional saturated aquifer material. Here, toluene was selected as a representative contaminant among the organic pollutants frequently detected in contaminated groundwater, and was used as a tracer for the natural attenuation and the fate of contaminants in the subsurface environment due to its high solubility and biodegradability. Their sorption type during transport was determined by comparing observed plumes with the simulated results, considering various sorption models. All these results were supported and described in detail by conducting numerical modeling. Furthermore, the nonlinear equilibrium adsorption model was more appropriate for estimating the retardation factor to explain the retardation of organic pollutants.

## 2. Materials and Methods

### 2.1. Two-Dimensional Plume Experiments

#### 2.1.1. Sand Box Model

Plume tests were performed in a two-dimensional physical aquifer model (Figure 1) which was constructed using polycarbonate with dimensions of 60 cm (L) × 30 cm (W) × 2 cm (H). Five ports with a diameter of 10 mm were positioned at 5-cm regular intervals on the left and right sides of the model to induce inflow and outflow during plume tests, respectively. Sampling ports of 171 units with a diameter of 7 mm were installed at the top of the aquifer model with a grid of 9 × 19 and capped by Teflon-coated rubber cap. To minimize air entrapment, the aquifer model standing in the longitudinal direction was filled with background liquid and then was uniformly packed with sandy soil, which mainly consisted of quartz (Jumunjin silica, Korea). Mechanical sieving of sandy material was performed using US Standard Sieves (Fisher Scientific, Waltham, MA, USA) No. 30 and No. 10 to obtain sand fractions (0.6~2.0 mm). Before experimental use, the sandy materials were washed using deionized water three times and autoclaved twice at 121 °C for 15 min to prevent any influence by other microorganisms. The bulk density and porosity of the sandy soil were determined to be 1.54 g/cm and 0.35, respectively. The sand was analyzed using an X-ray diffraction technique and was found to be mainly composed of quartz with very little organic carbon (<0.05%) [36].

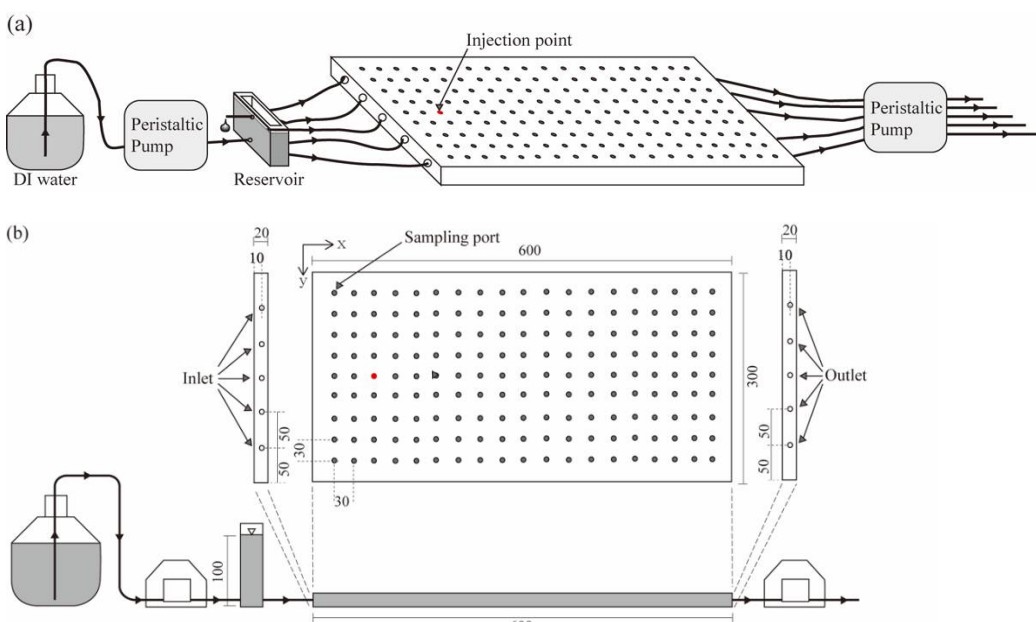

**Figure 1.** (**a**) Laboratory system for two-dimensional solute transport experiments; (**b**) layout of aquifer model (top view and front view).

2.1.2. Two-Dimensional Solute Transport Experiments

To test the fate and transport of organic pollutants in saturated media, solute transport experiments using toluene and KCl were performed in the sand box model. Steady-state flow conditions were imposed on the aquifer by applying a constant head ($\Delta h$ = 7 cm) and constant flux (Q = 22.68 mL/h) of DI water at the inflow side through a reservoir and a peristaltic pump, respectively. Once steady-state flow conditions were reached, tracer solutions were applied into the injection point of the aquifer model (x = 12 cm, y = 15 cm) for 6 min, using a syringe with injection rate $q_{in}$ = 10 mL/min. In the first case, a conservative tracer transport experiment was performed, using KCl to confirm the properties of solute transport through the advection dispersion process, except adsorption. This was carried out by injecting 60 mL of KCl solution at a concentration of 150 mg/L to investigate the dispersion of the solute. In the next case, 60 mL of toluene solution at a concentration of 200 mg/L were injected to investigate the effect of toluene sorption. Initial plumes of KCl and toluene were measured using separate additional tests. Samples were collected from the sampling port 24 h after the tracer injection. A minimum number of samples were collected using a 1-mL syringe in radial directions based on the expected center of the solute plume to minimize disturbance during sampling and to maximize the detection efficiency. KCl concentrations were analyzed using an electrical conductivity (EC) meter (Orion, Model: 130A, Hamburg, Germany). The toluene concentration was analyzed using HPLC (Young Lin, Seoul, Korea).

2.1.3. Moment Analysis for Solute Plume

Two-dimensional plumes can be characterized using moment analysis, which can express mass recovery and the center of mass. The travel distance of plumes were obtained as the distance from the injection point to the mass center of the plume. The mass center of plume was determined based on the calculation of the first raw moment, using moment analysis [37]. The coordinate location of the center of mass ($x_c$, $y_c$) was as follows:

$$x_c = \iint nC(x,y)x\,dxdy \Big/ \iint nC(x,y)\,dxdy \tag{1}$$

$$y_c = \iint nC(x,y)y\,dxdy \Big/ \iint nC(x,y)\,dxdy \tag{2}$$

where $C(x,y)$ is concentration field (M/L), $n$ is porosity (-) and $x,y$ are the coordinates (L).

The mass recovery of the KCl plume was determined based on the calculation of the observed mass using the inverse distance weighted interpolation method in GMS (version 10.3.4, Aquaveo, Prove, UT, USA). The mass recovery of tracer (Mr) was as follows:

$$\text{Mr} = \iint nC(x,y)\,dxdy \Big/ (C_0 \times t_0 \times q_{in}) \tag{3}$$

where $C_0$ is the concentration of the injected tracer (M/L$^3$), $t_0$ is injected time (T) and $q_{in}$ is injection flow rate (L$^3$/T).

*2.2. Modeling Water Flow and Solute Transport*

Two-dimensional solute transport was simulated using two different codes in the GMS (Aquaveo, Provo, UT, USA) environment. First, the flow part was executed using the modular three-dimensional finite-difference groundwater flow model (MODFLOW) code [38], the mathematical model simulating flow, using the partial-differential equation:

$$\frac{\partial}{\partial x}\left(K_{xx}\frac{\partial h}{\partial x}\right) + \frac{\partial}{\partial y}\left(K_{yy}\frac{\partial h}{\partial y}\right) + \frac{\partial}{\partial z}\left(K_{zz}\frac{\partial h}{\partial z}\right) - W = S_s\frac{\partial h}{\partial t} \tag{4}$$

where $K_{xx}$, $K_{yy}$ and $K_{zz}$ are the hydraulic conductivity along the x, y and z axes, respectively, which are assumed to be parallel to the major axis of the hydraulic conductivity (L/T); h is the potentiometric head (L); W is the volumetric flux per unit value, representing sources

and/or sinks of water (1/T); $S_S$ is the specific storage of porous material (1/L); and t is the time. The modeling domain was constructed with $62 \times 31$ grid cells and boundary conditions were as follows: the five source points (inlet ports) on the left side had constant head boundary conditions (h = 7 cm), the five sink points (outlet ports) on the right side had constant flux (4.536 cm$^3$/h), and the rest of the boundary was set as a no-flux boundary. The simulation conditions for two-dimensional water flow are listed in Table 1.

**Table 1.** Parameters used in the modeling of two-dimensional water flow.

| Parameter | Value | Source |
|---|---|---|
| Inflow: constant head | 7 cm | This study |
| Outflow: recharge source, W | 22.68 cm$^3$/h | This study |
| Hydraulic conductivity, $K_{xx}$, $K_{yy}$, $K_{zz}$ | 96.6 cm/h | [39] |
| Porosity of porous material, n | 0.35 | This study |
| Bulk density of porous material, $\rho_s$ | 1.57 g/cm$^3$ | This study |

Reactive solute transport was modeled using MT3D [40]. The general macroscopic equations describing the fate and transport of aqueous- and solid-phase species, respectively, in multi-dimensional saturated porous media are written as

$$\frac{\partial C}{\partial t} = \frac{\partial}{\partial x_i}\left(D_{ij}\frac{\partial C_k}{\partial x_j}\right) - \frac{\partial}{\partial x_i}(v_i C) + \frac{q_s}{\theta}C_s + r_c \tag{5}$$

where $C$ is the aqueous-phase concentration (M/L$^3$), $D_{ij}$ is the hydrodynamic dispersion coefficient (L$^2$/T), $v_i$ is the pore velocity (L/T), $\theta$ is the soil porosity, $q_s$ is the volumetric flux of water per unit volume of aquifer representing sources and sinks (1/T), $C_s$ is the concentration of source/sink (M/L$^3$), and $r_c$ represents the rate of all reactions that occur in the aqueous phase (ML$^3$/T).

The mobile species transport Equation (5) is first divided into four distinct equations: the advection equation, the dispersion equation, the source/sink-mixing equation, and the reaction equation. The reaction term in Equation (7) can be used to include the effect of geochemical reactions ($r_{cg}$) on contaminant fate and transport. The sorption of the solute can be described using two different models, assuming that the process follows equilibrium and kinetic irreversible sorption. These models were used independently to describe the solute sorption process during transport through saturated porous media.

$$r_{cg} = \frac{\rho_s}{\theta}\frac{\partial S_{eq}}{\partial t} + \frac{\rho_s}{\theta}\frac{\partial S_{irr}}{\partial t} \tag{6}$$

where the subscript $eq$ and $irr$ indicate equilibrium and kinetic irreversible sorption reaction, respectively. There are three types of equilibrium isotherms—linear, Freundlich and Langmuir in general. The sorption process in the linear isotherm can be represented as

$$\frac{\partial S_{eq}}{\partial t} = K_d\frac{\partial C_k}{\partial t} \tag{7}$$

where $K_d$ is the distribution coefficient (L$^3$/M). Equilibrium sorption isotherms are generally incorporated into the transport model through the use of the retardation factor (R). The retardation factor can be represented as

$$R = 1 + \frac{\rho_s}{\theta}K_d \tag{8}$$

where $\rho_s$ is the dry bulk density of soil (M/L$^3$). The sorption process in the Freundlich isotherm can be represented as

$$\frac{\partial S_{eq}}{\partial t} = K_f \left( \frac{\partial C_k}{\partial t} \right)^a \tag{9}$$

$$R = 1 + \frac{\rho_s}{\theta} a K_f C_k{}^{a-1} \tag{10}$$

where $K_f$ is the Freundlich constant ((L$^3$/M)$^a$) and $a$ is the Freundlich exponent ($-$). The sorption process in the Langmuir isotherm can be represented as

$$\frac{\partial S_{eq}}{\partial t} = \frac{K_l b \frac{\partial C_k}{\partial t}}{1 + K_l \frac{\partial C_k}{\partial t}} \tag{11}$$

$$R = 1 + \frac{\rho_s}{\theta} \left[ \frac{K_l b}{(1 + K_l C_k)^2} \right] \tag{12}$$

where $K_1$ is the Langmuir constant (L$^3$/M), and $b$ is the total concentration of sorption sites available (M/M). The sorption process in a kinetic irreversible site can be represented as

$$\frac{\rho_s}{\theta} \frac{\partial S_{irr}}{\partial t} = k_{irr} C_k \tag{13}$$

where $k_{irr}$ is (1/T).

### 2.3. Optimization of Parameters

The dispersion coefficient and sorption parameters were optimized through the comparison of the observed plume with those simulated. The solute concentration function is denoted as $C = f(X,t;p)$, where the vector $p$ represents the parameters and $X$ is the location of observation $(x,y)$. To optimize the dispersion coefficient, the observed and the simulated KCl plumes were compared, and the parameter $p = (D_x, D_y)$. To optimize the sorption coefficients, the observed and the simulated toluene plumes were compared, where the parameter $p = (k_{irr})$ was used for irreversible kinetic sorption, $p = (K_d)$ for a linear isotherm, $p = (K_f, a)$ for a Freundlich isotherm, and $p = (K_l, b)$ for a Langmuir sorption isotherm. The parameter domain was constrained so that $\Omega = [a_1,b_1]$ for single-parameter models and $\Omega = [a_1,b_1] \times [a_2,b_2]$ for two-parameter models, where the values of $a_n$ and $b_n$ are the minimum and maximum boundaries, respectively. For $m$ observation data at time t, $E = \{(X_i, C_i) | i = 1, \ldots, m\}$ is denoted as the set of the data. Then, the error function is defined as

$$e(p) = \sum_{i=1}^{m} (C_i - f(X_i; p_i))^2 \tag{14}$$

The discretized domain (DD) algorithm was used to determine the best parameters that could minimize $e(p)$ for given parameter domain $\Omega$ [41]. The boundaries and intervals used for the optimization algorithm are listed in Table 2. For the estimation of parameters with high resolution, the DD algorithm was repeatedly used in the finer mesh around the optimal point found by the preceding DD algorithm. For discretization of the parameter domain of single- parameter models, let $N_1$ be positive integers, let

$$\Omega = \left\{ x_i \middle| \log(x_i) = a_1 + \frac{i}{N_1} (b_1 - a_1), \ 0 \leq i \leq N_1 \right\} \tag{15}$$

be the set of nodes for preceding optimization, and let

$$\Omega = \left\{ x_i \middle| x_i = a_1 + \frac{i}{N_1} (b_1 - a_1), \ 0 \leq i \leq N_1 \right\} \tag{16}$$

be the set of nodes for the final optimization. For the discretization of the parameter domain of two-parameter parameters models, let $N_1$ and $N_2$ be positive integers, let

$$\Omega = \left\{ (x_i, y_j) \,\middle|\, \log(x_i) = a_1 + \frac{i}{N_1}(b_1 - a_1),\ \log(y_i) = a_2 + \frac{j}{N_2}(b_2 - a_2),\ 0 \le i \le N_1, 0 \le j \le N_2 \right\} \quad (17)$$

be the set of mesh grids for preceding optimization, and let

$$\Omega = \left\{ (x_i, y_j) \,\middle|\, x_i = a_1 + \frac{i}{N_1}(b_1 - a_1),\ y_i = a_2 + \frac{j}{N_2}(b_2 - a_2),\ 0 \le i \le N_1, 0 \le j \le N_2 \right\} \quad (18)$$

be the set of mesh grids for the final optimization. Then the minimum error can be approximated by min $e(p)$ for $p \in \Omega$ and values corresponding to this error can be chosen to be the best parameter set.

**Table 2.** Boundary values and intervals of parameters used for optimization algorithm.

| | Irreversible $k_{irr}$ (1/h) | Linear $K_d$ (cm³/mg) | Freundlich $K_f$ (cm³/mg) | $a$ (-) | Langmuir $K_l$ (cm³/mg) | $b$ (mg/mg) |
|---|---|---|---|---|---|---|
| Boundary | $(1.0 \times 10^{-5}, 1)$ | $(1.0 \times 10^{-9}, 1)$ | $(1.0 \times 10^{-6}, 1.0 \times 10^{-3})$ | $(1.0 \times 10^{-9}, 1)$ | $(1, 1.0 \times 10^{4})$ | $(1.0 \times 10^{-5}, 1)$ |
| Ratio [1] | 10 | 10 | 10 | 10 | 10 | 10 |
| Boundary | $(0.001, 0.2)$ | $(1.0 \times 10^{-8}, 1.0 \times 10^{-3})$ | $(1.0 \times 10^{-5}, 2.0 \times 10^{-4})$ | $(0.001, 0.1)$ | $(2300, 2500)$ | $(5.7 \times 10^{-4}, 5.9 \times 10^{-4})$ |
| Interval | 0.01 | 0.0001 | 0.0001 | 0.01 | 2400 | 0.00058 |

[1] At the first optimization, the parameter domain was discretized in a logarithmic scale.

## 3. Results and Discussion

### 3.1. Two-Dimensional Transport of Toluene and KCl Plumes

Observed plumes of KCl at 24 h after tracer injection are shown in Figure 2a. The results of moment analysis are listed in Table 3. The calculated mass center of the KCl plumes was located at $(x,y) = (37.65, 14.72)$. The travel distance from the injection point to the mass center of the KCl plumes was 25.65 cm. The average linear velocity $(v_x)$ was obtained as 1.069 cm/h. The mass recovery of KCl was 100.4%, respectively. The KCl plume at 24 h showed nearly a circular pattern with little distortion (Figure 2a).

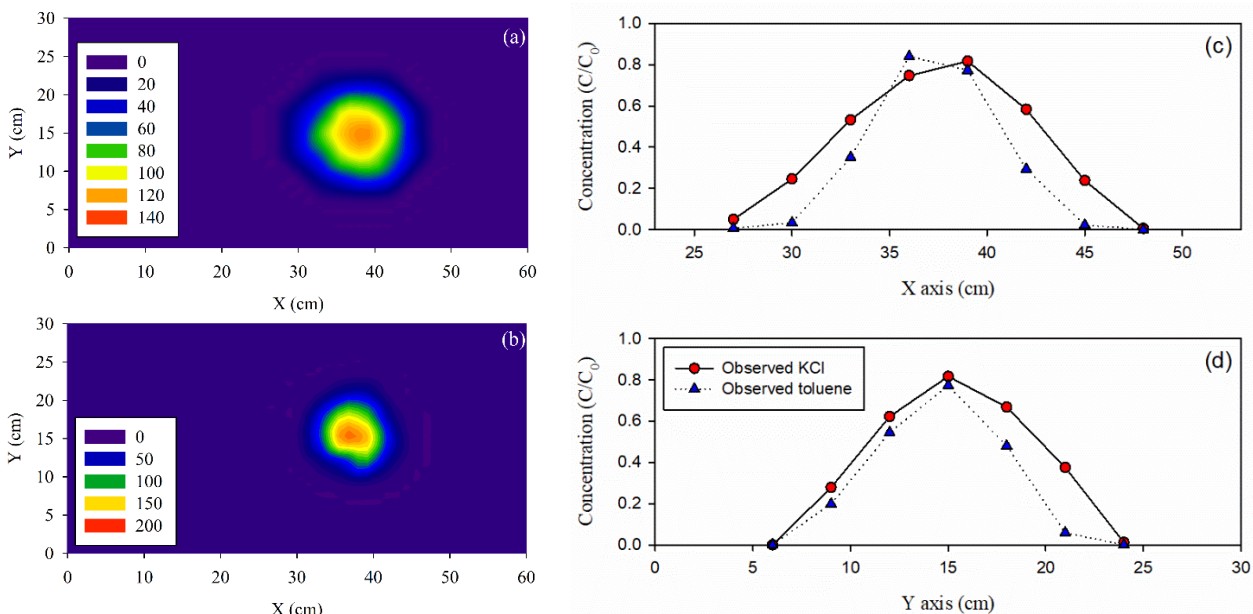

**Figure 2.** Observed (**a**) KCl and (**b**) toluene plumes at 24 h (unit of contour mg/L). Concentration distribution of toluene and KCl plume: (**c**) longitudinal direction (at y = 15 cm) and (**d**) transverse direction (at x = 39 cm).

**Table 3.** Mass center and mass recovery of observed and modeled KCl and toluene plumes obtained through moment analysis.

| Cases | KCl (mg/L) | Toluene (mg/L) | Mass Center (x,y) | Velocity (cm/h) | Mass Recovery (%) | Remark |
|---|---|---|---|---|---|---|
| K0 | 150 | - | (11, 15) | - | - | Obs. at 0 h |
| K24 | 150 | - | (37.65, 14.72) | 1.069 | 100.4 | Obs. at 24 h |
| T24 | - | 200 | (37.25, 14.35) | 1.052 | 60.2 | Obs. at 24 h |
| K0m | 150 | - | (11.81, 15.05) | - | - | Model at 0 h |
| K24m | 150 | - | (37.65, 14.72) | 1.047 | 100 | Model at 24 h |

The observed plumes of toluene at 24 h are shown in Figure 2b. The toluene peak depicted in 2c slightly precedes the KCl peak in the longitudinal direction, whereas in the transverse direction the peak distances are similar. The calculated mass center of the toluene plumes was located at (37.25, 14.35), with an average linear velocity ($v_x$) of 1.052 cm/h, similar to that of the KCl plume. This suggests that there was no retardation during the transport of toluene, even though the peak of toluene appeared to be slightly ahead of the peak of KCl. In contrast, the plume area of toluene was smaller than that of KCl. The mass of toluene was reduced due to the sorption of toluene onto the sandy soil during transport. The mass recovery of toluene was 60.2%, indicating about 40% toluene attenuation. Figure 3a,b shows a cross-section of the contaminant plume on the x and y axes, respectively. The peaks of the KCl and toluene plumes appear at the same relative concentration, whereas decreases in toluene concentrations were observed significantly at the edges of plumes. The attenuation of BTC without retardation could be simulated using first-order irreversible sorption, but these simulated BTCs could not explain the significant reduction in the solute concentration at the edge of plume [27–29]. This behavior is very different from what was reported in previous studies. The retarded transport observed in the several field tests did not occur in this study [33]. The significant concentration reduction at the boundary, rather than at the peak, is far from the characteristics of irreversible adsorption observed in the one-dimensional column test [27–29].

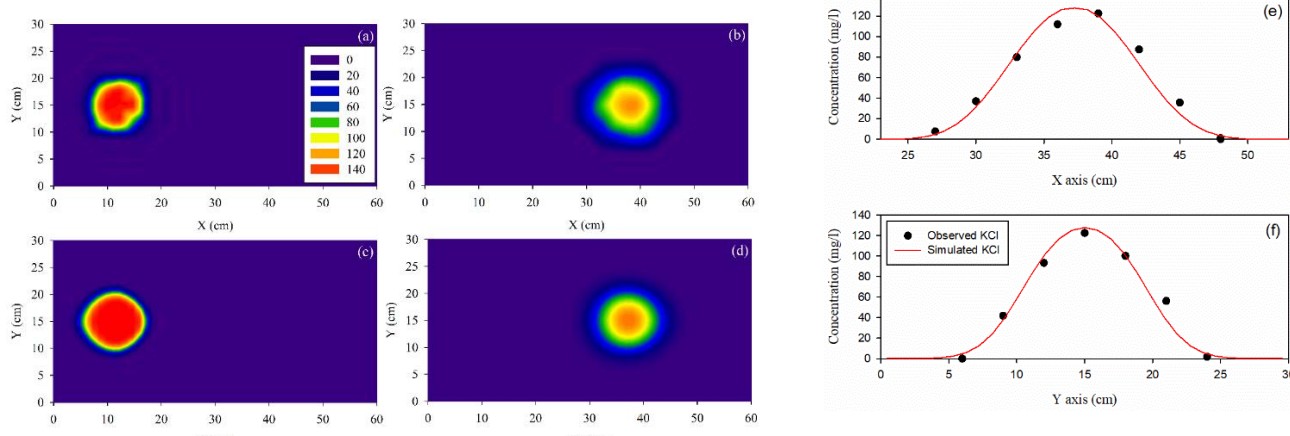

**Figure 3.** KCl plumes (unit of contour mg/L) observed (**a**) at the initial stage, (**b**) after 24 h; the simulated plumes (**c**) at the initial stage and (**d**) after 24 h; and (**e,f**) cross-section of concentration along the x- and y- axes after 24 h.

### 3.2. Modeling Advection and Dispersion in Two-Dimensional Sand Box

Numerical modeling for the two-dimensional water flow is depicted in Figure S1. Figure S1a,b shows equipotential lines during tracer injection and after tracer injection. During the tracer injection, the equipotential lines spread radially from the injection point. After tracer injection, the equipotential lines were parallel to the y axis and the flow in the x axis prevailed. The two-dimensional transport of KCl was modeled numerically, and the

dispersivity was obtained through comparison with the observed results. The optimized KCl plume is shown in Figure 3. The mass centers of the observed and the modeled KCl plumes were similarly calculated at 0 and 24 h. This indicates the successful simulation of the advective process. The moment analysis for the modeled plume showed that the average linear velocity (1.047 cm/h) is similar to that of the observed plume (1.067 cm/h). The dispersivity of KCl was estimated, with $\alpha_L$ = 0.141 cm, $\alpha_T/\alpha_L$ = 0.62. The obtained dispersivity was used as the dispersivity of toluene.

*3.3. Appropriate Model of Toluene Sorption during Transport through Saturated Sand*
3.3.1. Suitability of Linear Equilibrium and Nonequilibrium Irreversible Models

In the previous field-scale injection test for organic contaminant, a different degree of retarded transport was observed with respect to their hydrophobicity. This retardation could be explained by means of an equilibrium approach, using a linear sorption isotherm normalized using the soil organic carbon fraction. Several studies conducted on laboratory-scale tests have reported the retardation of organic contaminants using one- or two-dimensional experiment [30–32]. Several studies reported only the attenuation of organic contaminants without retardation, thus describing an irreversible type of sorption kinetics. Here, we tried to describe the sorption of a toluene plume using these sorption models. Using the parameter optimization algorithm, the error functions of the parameter domains are presented in Figure 4. The optimized parameters are tabulated in Table 4, and the optimized simulations of toluene plumes using each sorption model are compared with the observed results in Figure 5. The cross-sections of the toluene plumes along with the longitudinal and transverse directions are compared in Figure 6.

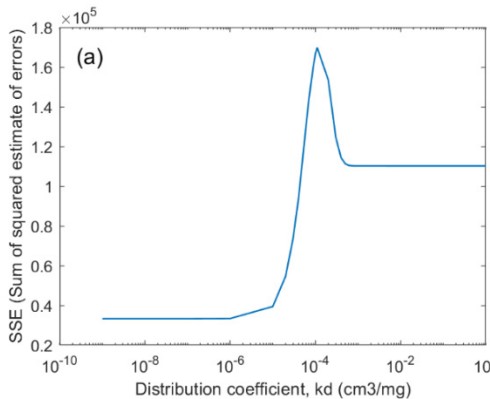 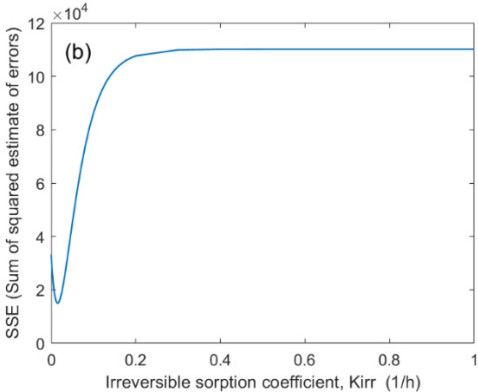

**Figure 4.** Goodness of fit in the simulations of toluene transport using (**a**) the linear isotherm model, (**b**) the kinetic-irreversible model. The curve shows the change in SSE (sum of squared error) according to the value of the adsorption parameter.

**Table 4.** Sorption parameters estimated by fitting sorption models to the observed plume of toluene.

| Model | Parameter | | | | | | SSE | $R^2$ | Mass Recovery (%) |
|---|---|---|---|---|---|---|---|---|---|
| | $k_{irr}$ (1/h) | $K_d$ (cm$^3$/mg) | $k_f$ (cm$^3$/mg) [a] | a (-) | $k_l$ (cm$^3$/mg) | b (mg/mg) | | | |
| Kinetic-irr | $1.5 \times 10^{-2}$ | - | - | - | - | - | $1.50 \times 10^4$ | 0.82 | 69.86 |
| Linear | - | $1.0 \times 10^{-8}$ | - | - | - | - | $3.33 \times 10^4$ | 0.60 | 99.99 |
| Freundlich | - | - | $7.76 \times 10^{-4}$ | $9.1 \times 10^{-4}$ | - | - | $2.25 \times 10^4$ | 0.73 | 73.73 |
| Langmuir | - | - | - | - | $2.48 \times 10^3$ | $5.72 \times 10^{-4}$ | $1.20 \times 10^4$ | 0.86 | 66.64 |

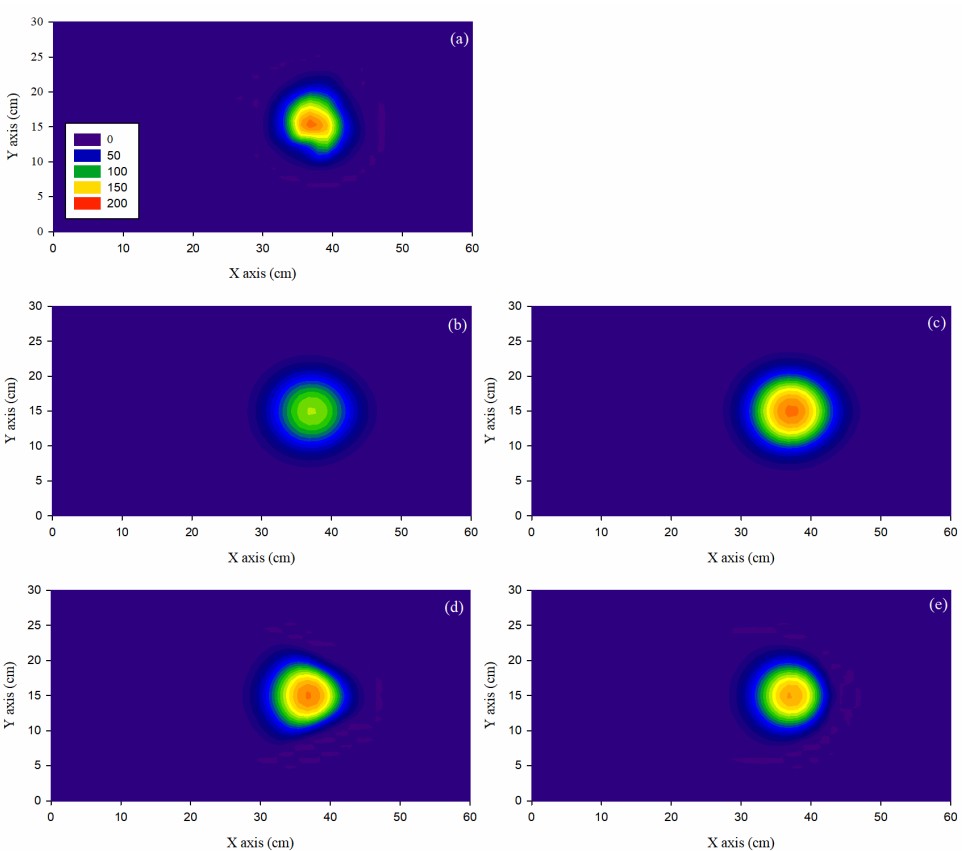

**Figure 5.** (**a**) Observed toluene plume, and the simulated using various sorption models (unit of contour mg/L): (**b**) kinetic-irreversible, (**c**) linear isotherm, (**d**) Freundlich isotherm and (**e**) Langmuir isotherm models.

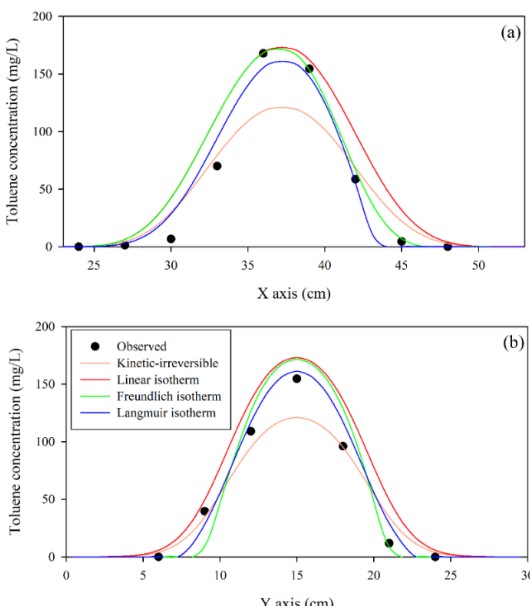

**Figure 6.** Concentration distribution of toluene plume: (**a**) x-axis direction (y = 15 cm); (**b**) y-axis direction (x = 39 cm).

We estimated the optimal sorption parameters of linear sorption isotherm, $K_d$, and kinetic irreversible sorption rate, $k_{irr}$. The error function in the parameter domain of $K_d$ is shown in Figure 4a. The local minimum is not unique, so the error function minimized at

below $1.0 \times 10^{-8}$ (cm$^3$/mg). At $K_d = 1.0 \times 10^{-8}$ cm$^3$/mg, the simulated toluene plume was not attenuated at all (mass recovery = 99.99 %), and the optimization results could not explain the observations ($R^2 = 0.60$). The comparison of the observed plume with the simulation also showed significant discrepancies (Figure 5c). This suggests that the linear sorption isotherms may not be adequate for the description of plume transport in the laboratory, even though it may be able to could describe the retardation during field-scale transport. This is a different result from the previous studies that have explained the behavior of organic pollution in a field conditions using with a linear sorption isotherm. This difference is thought to be caused by the difference between the organic matter content and the degree of contamination in the soil used in the experiment. The heterogeneous soil in field conditions contains clay and levels of natural organic matters that are significantly higher than $f_{oc}$. These soil conditions correspond corresponds to the isotherm conditions, with a large maximum adsorption amount. This, so that it can be interpreted as a situation in which soil was exposed to pollution at a level lower than the capacity of the soil. have. In this case, it is effective to explain the behavior of organic pollution through linear adsorption characteristics. On the contrary, in this experiment, soil with a minimal content of organic matter content was used through washing, which could that it can be interpreted as a situation in which the soil was exposed to a relatively high concentration of pollution due to its low sorption capacity. In such an environment, the equilibrium state can appear to be a phenomenon in which the pollutants are irreversibly removed. For this reason, it can be theorized that the characteristics of irreversible adsorption were observed in a one-dimensional experiment in which a high concentration of contamination was imposed, and that the same phenomenon can be explained by nonlinear adsorption with a low maximum adsorption amount.

The error function in the parameter domain of $k_{irr}$ is shown in Figure 4b. The unique local minimum of error function was found at $k_{irr} = 1.5 \times 10^{-2}$ /h. The simulated plume showed a high coefficient of determination ($R^2 = 0.82$) and the mass recovery (69.86%) was very similar to that observed for toluene (60.2%). Because the kinetic-irreversible model cannot induce retardation, the observe toluene plume was not retarded; the measured plume and the modeled were located in a similar position (Figure 5b). It is possible to explain the attenuation of toluene using irreversible sorption, as reported in some one-dimensional column studies. However, the peak concentration of the modeled plume was lower than that of the observed plume, and the modeled plume area was larger than the observed area (Figure 6).

### 3.3.2. Suitability of Non-Linear Equilibrium Models

The above results indicate that the two most commonly used models to describe the sorption of organic contaminants are insufficient to describe laboratory-scale two-dimensional transport characteristics. In the laboratory experiments, no retardation was observed, and the irreversible adsorption was explained by excessive contaminant removal. Conversely, we expected that the model of the strong adsorption of limited amounts of organic pollutants would be able to explain the adsorption characteristics of organic contaminants during transportation. These properties are well known as the properties of the nonlinear sorption isotherm. Therefore, the following analyses were conducted to determine whether the nonlinear equilibrium adsorption model is suitable for the description of the adsorption characteristics during the transport of two-dimensional toluene.

To select the type of sorption that is able to best represent the fate of toluene, two nonlinear equilibrium models (Freundlich isotherm and Langmuir isotherm) were applied to simulate toluene transport. We estimated the optimal sorption parameters of the Freundlich sorption isotherm, $K_f$ and $\alpha$. The error function in the parameter domain of $K_f$ and $\alpha$ is shown in Figure 7a. The optimization the algorithm was carried out by repeatedly reducing the parameter domain to find the best parameter set (Figure S2). The surface of the

error function was plotted on the parameter space with sufficient range for global optimization. The error function was minimized at $K_f = 7.64 \times 10^{-4}$ (cm$^3$/mg), $\alpha = 9.10 \times 10^{-4}$ (-) (Table 3).

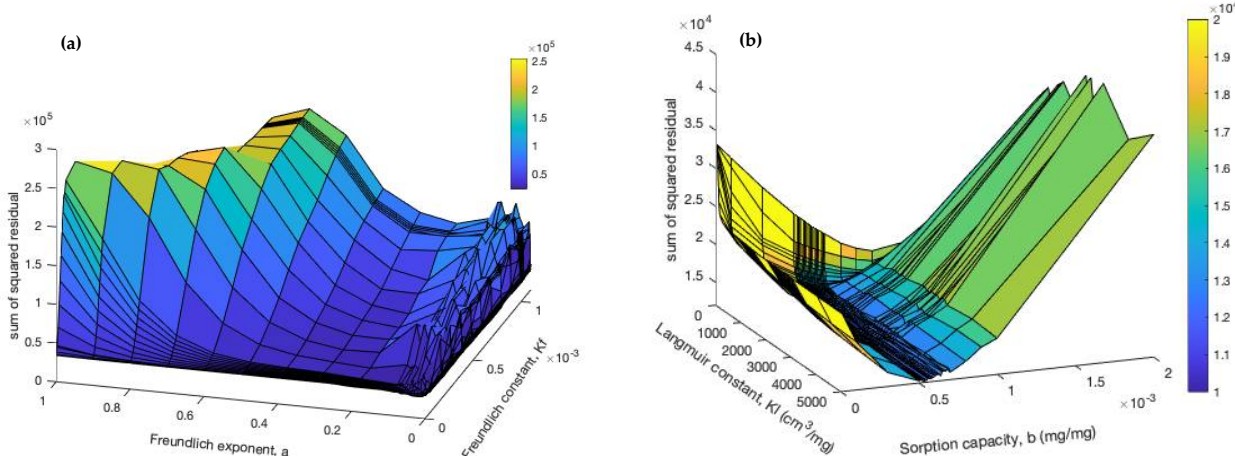

**Figure 7.** Goodness of fit in the simulation of toluene transport: (**a**) Freundlich isotherm model; (**b**) Langmuir isotherm model.

The plume simulated using the Freundlich isotherm models showed similar trends in terms of plume locations and areas with the observed results, and the sharpening of the solute front was also similarly explained (Figure 6). However, the mass recovery (73.73%) was evaluated higher than the observed results (60.2%) and the parameter optimization results were not able to explain the observations sufficiently ($R^2 = 0.73$).

Finally, we estimated the optimal sorption parameters of the Langmuir sorption isotherm, $K_l$ and $\beta$. The error function in the parameter domain of $K_l$ and $\beta$ is shown in Figure 7b. The optimization algorithm was carried out by repeatedly reducing the parameter domain to find the best parameter set (Figure S3). The change in the error function was sensitive to the sorption capacity ($\beta$), resulting in a valley with low error at $\beta = 5.0 \times 10^{-3}$ to $6.0 \times 10^{-3}$ on the surface of the error function. This indicates that the limited amount of the sorption site it is the most sensitive property in controlling the fate of toluene during transport. The error function was minimized at $K_l = 2.48 \times 10^3$ (cm$^3$/mg) and $\beta = 5.72 \times 10^{-4}$ (mg/mg) (Table 3). The Langmuir isotherm model explained the transport of toluene with a high goodness of fit ($R^2 = 0.86$) and showed a similar plume location, shape and mass reduction (mass recovery = 66.64 %) to the observed results. The simulated plume was also able to explain the sharpening of the solute front (Figure 6).

In Figure 6, the observed distribution of the toluene plume and the distribution simulated using optimized sorption parameters are compared using cross-sectional views in longitudinal and transverse directions. The irreversible kinetic sorption model showed lower peak concentrations, whereas the linear isotherm model showed higher distributions than the observed concentrations. The Freundlich isotherm and the Langmuir isotherm showed lower concentration distributions at the edges of the contaminant plume than the irreversible kinetic sorption model and the linear isotherm model. Among these models, the Langmuir isotherm model showed the curve that was closest to the observed values.

The smaller size of the toluene plume compared to the size of the conservative tracer plume indicates that the sorption is faster at the flange of the plume. This indicates that in the distribution of toluene concentration by dispersion, rapid sorption occurs at the flange (low concentration) and less sorption occurs at the center (high concentration). In other words, the quartz sand used in this study places a limitation on the sorption capacity of toluene. Barry et al. (2002) reported that the Langmuir model is applicable to cases in which sorption is limited to a finite capacity that is being represented [42]. Many studies on the sorption of various dissolved organic compounds onto the soil have reported that the Freundlich and linear models are more suitable than the Langmuir model [43–45]. This is because all models are similar to the linear model at low concentrations [46]. Ball and

Roberts (1991) reported that nonlinear isotherms (Langmuir and Freundlich models) of tetrachloroethene (PCE) and 1,2,4,5-tetrachlorobenzene (TeCBz) on sandy aquifer solids fit the entire range of data much better than the simple linear relationship did [47]. At the low concentrations (<50 mg/L) relevant to the rate studies [47,48] and field experiments [49] the isotherm data appeared to be more linear. To simulate the transport of high-concentration hydrocarbons, as in this study, the use of models with limited sorption capacity should be considered.

The above results indicate that the nonlinear adsorption characteristics were certainly valid for explaining the transport of organic contaminants. The sorption isotherms for the optimized parameters, presented in Figure 8, show that the two nonlinear isotherms have a limited adsorption capacity to solids. This might be related to the soil conditions, which had a low organic content. The sandy soil used in this experiment was mainly composed of quartz, and because organic matter is removed by washing, it is considered to have relatively fewer adsorption sites than general soil. Therefore, the nonlinear adsorption characteristics with limited adsorption sites could explain the experimental results well. The Freundlich isotherm is a model in which the amount of adsorption increases nonlinearly with an increasing liquid concentration. However, it was expressed as having a limited adsorption site in the optimized simulation. This seems to be because the optimized Freundlich exponent, b (−), was too small (value = 0.00091), and so it converged to the zero-order reaction.

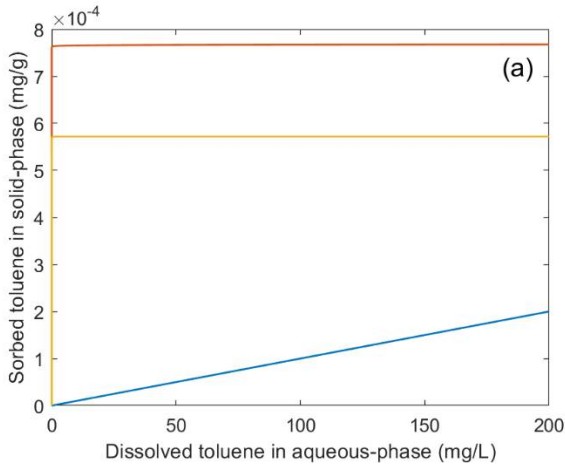
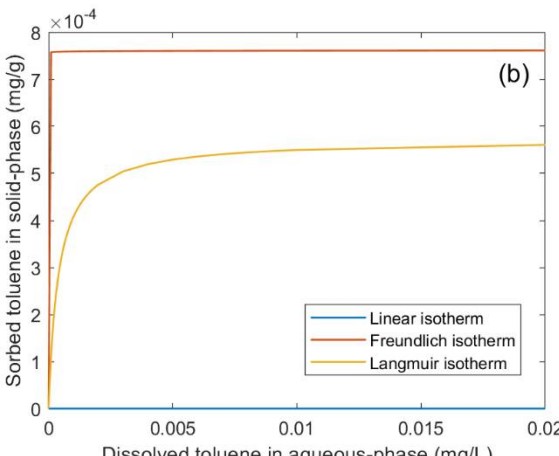

**Figure 8.** Estimated sorption isotherm curve in the simulation of toluene transport: (**a**) toluene concentration = 0~200 mg/L; (**b**) toluene concentration = 0~0.02 mg/L.

The Langmuir isotherm, shown in Figure 8b, indicates that the soil toluene adsorption capacity was already saturated at toluene concentrations above 0.005 mg/L. It can be inferred that the reactive site of the soil surface with a small capacity is easily saturated by a low concentration of the contaminant through hydrophobic adsorption, and no further adsorption proceeds. This reduces the concentration of organic pollutants that are more pronounced in the areas with low concentrations, such as in the vicinity of a plume. This may be attributed to the decrease in the size of the toluene plume during transport.

## 4. Conclusions

The transport of an organic contaminant through sandy material was investigated using an areal two-dimensional saturated aquifer model. In the toluene transport experiment, the toluene plume showed a decrease in mass, as well as plume size. The shape of the toluene plume could be explained using the nonlinear sorption model, especially the Langmuir isotherm model. We conclude that for the retarded and attenuated transport of organic pollutants, it is appropriate to use a transport model that considers nonlinear isothermal adsorption. The results of this study also suggest a methodology for evaluating

the fate and transport characteristics of organic pollutants in aquifers. The adsorption of organic pollutants—which is sensitively affected by organic matter contained in small amounts in the soil—is difficult to distinguish from irreversible adsorption characteristics through one-dimensional laboratory experiments. In this case, two-dimensional transport experiments may be a promising alternative. In particular, two-dimensional experiments are more appropriate than one-dimensional column experiments in identifying the adsorption characteristics that occur during transport in saturated aquifers. The short pulse injection in the 2D sand box was able to monitor the transport, dispersion and attenuation of contaminants at the margin, as well as the center of the plume. Therefore this study can supply an experimental approach to assessing the natural attenuation of organic contaminants in a subsurface environment which has a high contamination level in heterogeneous media, affected by various (a)biotic processes.

**Supplementary Materials:** The following are available online at https://www.mdpi.com/article/10.3390/w13111557/s1, Figure S1: Result of flow modeling using MODFLOW code (unit of contour, cm): (a) injection time ($0 < t < t_0$), (b) after injection ($t > t_0$), Figure S2: Modeling procedure for the Freundlich sorption isotherm (Freundlich constant ($K_f$), Freundlich exponent ($a$)): (a) $K_f = 1.0 \times 10^{-6} \sim 1.1 \times 10^{-3}$; $a = 1.0 \times 10^{-8} \sim 1.0$; (b) $K_f = 1.0 \times 10^{-5} \sim 2.0 \times 10^{-4}$; $a = 1.0 \times 10^{-3} \sim 1.0 \times 10^{-1}$. Figure S3: Modeling procedure for the Langmuir sorption isotherm (Langmuir constant ($K_l$); total concentration of sorption sites available ($b$)): (a) $K_l = 1.0 \sim 1.0 \times 10^4$; $b = 1.0 \times 10^{-5} \sim 1.0$; (b) SSR = $1.2 \times 10^4 \sim 1.25 \times 10^4$, (c) $K_l = 2.35 \times 10^3 \sim 2.25 \times 10^3$; $b = 5.7 \times 10^{-4} \sim 5.9 \times 10^{-4}$.

**Author Contributions:** Conceptualization, S.L. and J.-W.C.; methodology, S.-G.L.; software, S.-G.L.; validation, S.L.; formal analysis, S.L.; investigation, S.-G.L.; writing—original draft preparation, S.-G.L.; writing—review and editing, J.-W.C.; supervision, J.-W.C.; project administration, S.L. All authors have read and agreed to the published version of the manuscript. Please turn to the CRediT taxonomy for the explanation of terms.

**Funding:** This research was funded by Ministry of Environment (MOE), grant number 2018002470002 and Korea Ministry of Science and ICT (grant number NRF-2017R1C1B3009500).

**Data Availability Statement:** Not applicable.

**Acknowledgments:** This work was supported by the Korea Environment Industry Institute (KEITI) through the Underground environmental pollution risk management technology development business Program, funded by the Ministry of Environment (MOE) (grant number: 2018002470002), and the National Research Foundation of Korea (NRF) grant funded by the Korea Ministry of Science and ICT (grant number NRF-2017R1C1B3009500).

**Conflicts of Interest:** The authors declare no conflict of interest.

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
