# Peer review of "Nonlinear Sorption of Organic Contaminant during Two-Dimensional Transport in Saturated Sand"

_water, doi:10.3390/w13111557_

Round 1

Reviewer 1 Report

This paper presents the case study of nonlinear sorption of organic contaminant using two-dimensional transport model in saturated sand. The goal of the work is well explained and authors provided a detailed explanation of the methodology used to study sorption processes in the simulated environment. The literature review is thorough and well documented. Conclusions are clear and are in accordance with the results of research. I have only few remarks and suggestions:

  1. English language is correct, however it is recommended to edit the paper before publication.
  2. In abstract, it is necessary to avoid repeating the word appropriateness too often, it is recommended to use some similar wording
  3. Line 90-92: incomprehensible sentence, it needs to be reformulated to be more understandable
  4. Authors are recommended to better explain why they selected toluene as a representative organic contaminant in this particular case study.
  5. Line 122-123: incomprehensible sentence, it needs to be reformulated to be more understandable
  6. Lines 152, 367: the part of the sentence: Bulleted list look like this… What is it?
  7. Line 263 – please explain in more detail the mass recovery of KCL. Mass recovery of 97,9% is not shown in the table 3.

Author Response

Thanks for your comments.

1. English language is correct, however it is recommended to edit the paper before publication.

: We have revised and edited again our manuscript as you comment.

2. In abstract, it is necessary to avoid repeating the word appropriateness too often, it is recommended to use some similar wording

: We have revised the abstract based on your recommendation. 

3. Line 90-92: incomprehensible sentence, it needs to be reformulated to be more understandable

: The sentence has been revised as follows 

"Previous studies were reported on the difference of retardation caused by non-linear and linear sorption using a one-dimensional column." 

4. Authors are recommended to better explain why they selected toluene as a representative organic contaminant in this particular case study.

: The sentence has been revised as follows 

"Here, toluene was selected as a representative contaminant among the organic pollutants frequently detected in the contaminated groundwater, which is used as a tracer for the natural attenuation and the fate of contaminants in subsurface environment due to its high solubility and biodegradability. "

5. Line 122-123: incomprehensible sentence, it needs to be reformulated to be more understandable

: The sentence has been revised as follows 

"Before experimental use, the sandy materials were washed using deionized water three times, and autoclaved twice at 121°C for 15 min to prevent any influence by other microorganisms. "

6. Lines 152, 367: the part of the sentence: Bulleted list look like this… What is it?

: These typos have been removed. 

7. Line 263 – please explain in more detail the mass recovery of KCL. Mass recovery of 97,9% is not shown in the table 3.

: This value was the result of a duplicate experiment that was excluded from this study because of its high similarity. This value have been removed. 

Reviewer 2 Report

The paper "Nonlinear sorption of organic contaminant during two-dimensional 2 transport in saturated sand " deals with the problem of contaminants fate in the environment, namely groundwater. 

The issue is interesting and has some implications and possibile applications for environment protection, including a particular specific interest for laboratory tests using column, boxes or tanks, aiming at studying transport and sorption behavior of a pollutant in a natural medium.

The introduction encompasses a detailed description of the effects and properties of organic contaminants, and is very well written, even in the part where experimental sketch is described to study sorption effects on transport.

I suggest to cite other papers where column tests are reported as they are very important as a starting point for your following discussion.

Even for the field experiments it is very important to accurately describe the state of the art as a beginning for the introduction of new considerations and contributions from your research.

Even when you talk about the equilibrium approach using partitioning coefficient, you should cite some paper about it.

In the materials and methods chapter, the description of the apparatus test is very accurate, but, in my opinion Figure 1 should be better replaced by a picture of the laboratory system used for the tests (or, better, you can mantain this scheme and add a picture).

This would help in the comprehension of the test apparatus.

I understand that the aquifer model is filled with soil, sandy soil for the precision; can you indicate more properties of this soil? (for example bulk density, porosity, grain diameter and so on? and more precise feature and characteristics of the alluvial aquifer from which it comes?

In the paper you indicate some values but it is not so clear of they refer to the original soil sampled in the field or to the portion you select for the aquifer model construction (0.6-2 mm).

I suggest (page 4) to add a simplified figure where the scheme of modflow initial conditions (with sinks, cells, boundary conditions) is reported and not only described.

The results and discussion chapter contains rather detailed description of the observed processes during the experiments. In my opinion you should discuss why the Toluene peak in the 2c slightly precedes the KCl peak ( in the longitudinal direction while in the transenne direction the peak distances are similar.

At page 8 you state that "The peaks of KCl and toluene plumes appear at the same relative concentration, while toluene concentrations decrease was observed significantly at the edges of plumes. This behavior is very different from what was reported in previous studies. The retarded transport observed in the several field tests was not occurred in this study [27]. And the significant concentration reduction at the boundary, not at the peak is far from the characteristics of irreversible adsorption in one dimensional column test". You should discuss this result, and indicate why you obtain this outcome respect to which is reported in the literature.

At page 10, in the Figure 4 caption, maybe it would be better clearly indicate that the curves are erf, for a better comprehension.

At page 11 you state "The Freundlich isotherm models showed similar plume locations and areas, but it was difficult to accurately simulate the reduction of mass. Although the simulated toluene plume was attenuated (mass recovery = 73.73 %), and the optimization results could not explain the observations (R2 = 0.73).": although I think that the sentence needs to be re-written in order to allow understanding of the exact meaning the author wanted to express, in my opinion it is also necessary to explain more in detail why the toluene mass recovery is so low in case of Freundlich isotherm.

Conclusions are synthetic and well express the core of the paper. The only comment I express is related to the goodness of the test apparatus to investigate the main point of your research. I'm not so sure that using a larger apparatus some effects (longitudinal and transversal dispersion, retardation factor) do not show diverse behaviour... Maybe you can briefly discuss it in this paper and, if you want, this can be a future development.

I attach some minor changes in the pdf file.

Author Response

Point 1. The introduction encompasses a detailed description of the effects and properties of organic contaminants, and is very well written, even in the part where experimental sketch is described to study sorption effects on transport.

I suggest to cite other papers where column tests are reported as they are very important as a starting point for your following discussion.

Even for the field experiments it is very important to accurately describe the state of the art as a beginning for the introduction of new considerations and contributions from your research.

Even when you talk about the equilibrium approach using partitioning coefficient, you should cite some paper about it.

Response 1: Thanks for your comment. As your comments, we cited previous researches as follows:

Numerous experimental and modeling studies have been performed in laboratory and/or field scales to examine the effect of sorption or organic contaminants onto aquifer materials and its consequences on their transport phenomena [14,23-33]. Especially, the column experiments have been frequently used to investigate the fate of organic contaminants [26]. Several laboratory column studies have shown the attenuation of organic contaminants during transport [27-29]. The retardation of organic pollutants have also been reported in one-dimensional column experiments as well as in the two-dimensional sand box test [30-32]. In several field studies, highly retarded transport trends were reported for hydrophobic compounds [33]. Long-term experiments at the site scale (Borden, Ontario) confirmed the occurrence of retardation along with attenuation, and this behavior could be successfully explained using linear equilibrium approach [14].

The equilibrium approach using partitioning coefficient (Kd = focKoc) is widely used to describe the sorption during transport of organic contaminant in aquifer [14,30-32] . However, the linear sorption property is not applicable in all parts. Previous studies were reported on the difference of retardation caused by non-linear and linear sorption using a one-dimensional column [34,35].” (Line 76 – 91)

Point 2. In the materials and methods chapter, the description of the apparatus test is very accurate, but, in my opinion Figure 1 should be better replaced by a picture of the laboratory system used for the tests (or, better, you can mantain this scheme and add a picture).

This would help in the comprehension of the test apparatus.

Response 2: Figure 1 is replaced with revised figure as your comment.

Point 3. I understand that the aquifer model is filled with soil, sandy soil for the precision; can you indicate more properties of this soil? (for example bulk density, porosity, grain diameter and so on? and more precise feature and characteristics of the alluvial aquifer from which it comes?

In the paper you indicate some values but it is not so clear of they refer to the original soil sampled in the field or to the portion you select for the aquifer model construction (0.6-2 mm).

Response 3: The soil used in this study was purchased from Jumunjin silica (Korea). Since we did not sample by ourself, we don’t have any soil property in field condition. The informations on the portion we select for the sandbox construction is described as follows:

Mechanical sieving of sandy material was performed using US Standard Sieves (Fisher Scientific, USA), No. 30 and No. 10 to obtain sand fractions (0.6 ~ 2.0 mm). Before experimental use, the sandy materials were washed using deionized water three times and autoclaved twice at 121°C for 15 min to prevent any influence by other microorganisms. The bulk density and porosity of the sandy soil were determined to be 1.54 g/cm and 0.35, respectively. The sand was analyzed by X-ray diffraction technique and was found to be mainly composed of quartz with very little organic carbon (<0.05 %) [36].” (Line129)

Point 4. I suggest (page 4) to add a simplified figure where the scheme of modflow initial conditions (with sinks, cells, boundary conditions) is reported and not only described.

Response 4: The location of source and sink points can be explained in the revised Figure 1. Referring the revised figure, MODFLOW initial condition can be sufficiently explained.

Point 5. The results and discussion chapter contains rather detailed description of the observed processes during the experiments. In my opinion you should discuss why the Toluene peak in the 2c slightly precedes the KCl peak ( in the longitudinal direction while in the transenne direction the peak distances are similar.

Response 5: We discussed this point as follows:

 “Observed plumes of toluene at 24 h are shown in Figure 2b. Toluene peak in the 2c slightly precedes the KCl peak in the longitudinal direction while in the transverse direction the peak distances are similar. Calculated the mass center of toluene plumes was located at (37.25, 14.35), with average linear velocity (vx) of 1.052 cm/h, similar with that of KCl plume. This suggests that there was no retardation during the transport of toluene even thought the peak of toluene appeared to be slightly ahead of the peak of KCl.” (Line 298~ 303)

Point 6.  At page 8 you state that "The peaks of KCl and toluene plumes appear at the same relative concentration, while toluene concentrations decrease was observed significantly at the edges of plumes. This behavior is very different from what was reported in previous studies. The retarded transport observed in the several field tests was not occurred in this study [27]. And the significant concentration reduction at the boundary, not at the peak is far from the characteristics of irreversible adsorption in one dimensional column test". You should discuss this result, and indicate why you obtain this outcome respect to which is reported in the literature.

Response 6: We added following sentence:

“The attenuation of BTC without retardation could be simulated using first order irreversible sorption, but this simulated BTCs could not explain the significant reduction of solute concentration at the edge of plume [27-29].” (Line 309-312)

Point 7. At page 10, in the Figure 4 caption, maybe it would be better clearly indicate that the curves are erf, for a better comprehension.

Response 7: Figure 4 caption corrected as follows:

Figure 4. Goodness of fit in the simulations of toluene transport using (a) linear isotherm model, (b) kinetic-irreversible model. The curve shows the change of SSE (sum of squared error) according to the value of the adsorption parameter.

Point 8. At page 11 you state "The Freundlich isotherm models showed similar plume locations and areas, but it was difficult to accurately simulate the reduction of mass. Although the simulated toluene plume was attenuated (mass recovery = 73.73 %), and the optimization results could not explain the observations (R= 0.73).": although I think that the sentence needs to be re-written in order to allow understanding of the exact meaning the author wanted to express, in my opinion it is also necessary to explain more in detail why the toluene mass recovery is so low in case of Freundlich isotherm.

Response 8: We corrected the paragraph as follows:

 “The plume simulated using The Freundlich isotherm models showed similar trends in plume locations and areas with the observed, and so the sharpening of the solute front was also similarly explained (Figure 6). However the mass recovery (73.73 %) was evaluated higher than the observed (60.2%) and the parameter optimization results could not explain the observations sufficiently (R2 = 0.73).” (Line 427-431)

Point 9. Conclusions are synthetic and well express the core of the paper. The only comment I express is related to the goodness of the test apparatus to investigate the main point of your research. I'm not so sure that using a larger apparatus some effects (longitudinal and transversal dispersion, retardation factor) do not show diverse behaviour... Maybe you can briefly discuss it in this paper and, if you want, this can be a future development.

Response 9: Thanks for your comment. We revise Conclusion as follows:

"The transport of organic contaminant through sandy material was investigated using the areal two-dimensional saturated aquifer model. In the toluene transport experiment, toluene plume showed decrease in mass as well as plume size. The shape of the toluene plume could be explained using nonlinear sorption model, especially Langmuir isotherm. We conclude that, for the retarded and attenuated transport of organic pollutants, it is appropriate to use a transport model that considers nonlinear isothermal adsorption. The results of this study also suggest a methodology for evaluating the fate and transport characteristics of organic pollutants in aquifers. Ad-sorption of organic pollutants sensitively affected by organic matter in the soil contained in small amounts is difficult to distinguish from irreversible adsorption characteristics through one-dimensional laboratory experiments. In this case, two-dimensional transport experiments may be a prominent alternative. Especially, the two-dimensional experi-ments are more appropriate than one-dimensional column experiments to identify the adsorption characteristics that occur during transport in saturated aquifers. The short pulse injection in 2D sand box could monitor the transport, dispersion and attenuation of contaminants at the margin as well as the center of the plume. Therefore this can supply an experimental approach to assess the natural attenuation of organic contaminant in the subsurface environment which has wide contamination level in heterogeneous media affected by various (a)biotic processes."

Reviewer 3 Report

water-1234484: Nonlinear sorption of organic contaminant during two-dimensional transport in saturated sand

This manuscript examines the transport of toluene in a two-dimensional bench scale, homogeneous, water saturated porous medium. Certainly, there are not too many experimental studies available in the literature contaminant transport in two- and three-dimensional systems. The manuscript is organized and written very well. The experimental procedures and data analysis are thoroughly presented. The figures are clear. Consequently, this reviewer recommends publication of this manuscript in Waterafter a revision.The following is a short list of suggestions that may improve this manuscript prior its publication:

(1) Lines 80-81 & 307-308: The statement that very few studies reported the retardation of organic pollutants is misleading. There are numerous studies that report retardation factors of organic pollutants (e.g., Water Resources Research, 36(7), 1687–1696, 2000; Water Research, 36, 3911–3918, 2002; Journal of Hazardous Materials, B128, 218-226, 2006).

(2) The readers will benefit to know that heterogeneous soil in the field cannot be treated as uniform and homogeneous. Actually, for field cases the retardation factor often is spatially variable due to the variability in the soil properties and foc. For such heterogeneous porous media, the retardation factor is not constant, but spatially variable and is known to yield increased spreading of the dissolved contaminant (Water Resources Research, 26(3), 437–446, 1990; Water Resources Research, 28(6), 1517–1529, 1992; Transport in Porous Media, 7(2), 163–185, 1992).

(3) It is unclear how many parameters are fitted. Are the fitted parameters unique? This may not be true if the number of fitted parameters is greater than 3.

(4) The authors need to clearly state in the revised manuscript why the work presented is needed.

(5) Will the results from this study be more realistic or more accurate if the experiments were conducted in a three-dimensional model aquifer?

Author Response

Thanks for your comments. 

Point 1) Lines 80-81 & 307-308: The statement that very few studies reported the retardation of organic pollutants is misleading. There are numerous studies that report retardation factors of organic pollutants (e.g., Water Resources Research, 36(7), 1687–1696, 2000; Water Research, 36, 3911–3918, 2002; Journal of Hazardous Materials, B128, 218-226, 2006).

Response 1. We appreciate for your comments. The above mentioned research are referred in our revised manuscript as follows.

“Several laboratory column studies have shown the attenuation of organic contaminants during transport [27-29]. The retardation of organic pollutants have also been reported in one-dimensional column experiments as well as in the two-dimensional sand box test [30-32].                          (Line 81-84 ).”

“Several studies conducted in laboratory scale test reported the retardation of organic contaminant using one or two dimensional experiment [30-32] (Line 310-311).”

Point 2) The readers will benefit to know that heterogeneous soil in the field cannot be treated as uniform and homogeneous. Actually, for field cases the retardation factor often is spatially variable due to the variability in the soil properties and foc. For such heterogeneous porous media, the retardation factor is not constant, but spatially variable and is known to yield increased spreading of the dissolved contaminant (Water Resources Research, 26(3), 437–446, 1990; Water Resources Research, 28(6), 1517–1529, 1992; Transport in Porous Media, 7(2), 163–185, 1992).

Response 2. Based on your comments, we added following sentence in Introduction.

“In field condition, heterogeneous soil properties such as organic matter further complicate the fate of organic contaminant [15-17]. (Line 64).”

Point 3) It is unclear how many parameters are fitted. Are the fitted parameters unique? This may not be true if the number of fitted parameters is greater than 3.

Response 3. The number of fitted parameters was not exceed two in the research.

Point 4) The authors need to clearly state in the revised manuscript why the work presented is needed.

Responce 4. We added the following sentences to present the purpose of the experimental cases.

“To test the fate and transport of organic pollutants in saturated media, a solute transport experiments using toluene and KCl were performed in the sand box model. Steady-state flow condition was imposed on the aquifer by applying a constant head (Δh=7 cm) and constant flux (Q= 22.68 ml/h) at the inflow side through a reservoir and a peristaltic pump respectively. Once the steady-state flow condition was reached, tracer solutions were applied into the injection point of the aquifer model (x =12 cm, y =15 cm) for 6 min using a syringe with injection rate qin=10 ml/min.At first case, conservative tracer transport experiment was performed using KCl to confirm the properties of solute transport through the advection dispersion process except adsorption. It was conducted by injecting 60 ml of KCl solution at concentration of 150 mg/l to investigate the dispersion of solute. In the next case, 60 ml of toluene solution of at concentration of 200 mg/l were injected to investigate the effect of toluene sorption. ….. (Line 141-) “

“Two-dimensional plume can be characterized by moment analysis that can express mass recovery and center of mass. (Line 163)”